# Dynamics and consequences of spliceosome E complex formation

**Joshua Donald Larson[1,2], Aaron A Hoskins[1,2]***

[1]Department of Biochemistry, University of Wisconsin-Madison, Madison, United States; [2]Biophysics Graduate Program, University of Wisconsin-Madison, Madison, United States

**Abstract** The spliceosome must identify the correct splice sites (SS) and branchsite (BS) used during splicing. E complex is the earliest spliceosome precursor in which the 5' SS and BS are defined. Definition occurs by U1 small nuclear ribonucleoprotein (snRNP) binding the 5' SS and recognition of the BS by the E complex protein (ECP) branchpoint bridging protein (BBP). We have used single molecule fluorescence to study *Saccharomyces cerevisiae* U1 and BBP interactions with RNAs. E complex is dynamic and permits frequent redefinition of the 5' SS and BS. BBP influences U1 binding at the 5' SS by promoting long-lived complex formation. ECPs facilitate U1 association with RNAs with weak 5' SS and prevent U1 accumulation on RNAs containing hyperstabilized 5' SS. The data reveal a mechanism for how U1 binds the 5' SS and suggest that E complex harnesses this mechanism to stimulate recruitment and retention of U1 on introns.

DOI: https://doi.org/10.7554/eLife.27592.001

## Introduction

Intron removal by the spliceosome is an essential step in precursor messenger RNA (pre-mRNA) processing during eukaryotic gene expression. The spliceosome is composed of a number of subunits including the five small nuclear ribonucleoproteins (the U1, U2, U4, U5, and U6 snRNPs), each containing a U-rich small nuclear RNA (snRNA) and several snRNP-specific proteins (*Wahl et al., 2009*). Spliceosomes assemble stepwise on pre-mRNAs via a partially ordered pathway to correctly identify the boundaries of the intron and carry out the splicing reaction (*Hoskins et al., 2011*; *Shcherbakova et al., 2013*; *Wahl et al., 2009*). Assembly often begins with ATP-independent binding of the U1 snRNP to the 5' splice site (5' SS) to form the spliceosome E complex (*Michaud and Reed, 1991*; *Mount et al., 1983*; *Séraphin et al., 1988*; *Zhuang and Weiner, 1986*). Along with U1, E complex also contains a number of other splicing factors (E complex proteins, ECPs) associated with other locations on the transcript (*Figure 1A*). The nuclear cap binding complex (CBC) is bound to the 5' cap of the pre-mRNA (*Colot et al., 1996*) while additional ECPs associate with the intron branchsite (BS; BBP/Mud2 in the yeast or SF1/U2AF1 and 2 in humans) (*Abovich and Rosbash, 1997*; *Abovich et al., 1994*; *Bennett et al., 1992*; *Berglund et al., 1997*; *Zamore et al., 1992*; *Zhang and Rosbash, 1999*).

E complex is the earliest spliceosome assembly intermediate conserved between yeast and humans that also marks the approximate locations for both 5' SS cleavage and the branchpoint adenosine responsible for carrying out the cleavage reaction. Ample evidence has shown that E complexes are competent intermediates for subsequent spliceosome assembly (*Seraphin and Rosbash, 1989*; *Séraphin et al., 1988*; *Michaud and Reed, 1991*). In yeast, the E complex has been referred to as the commitment complex (CC) since at least a subset of the complexes being formed are resistant to addition of competitor RNA and can be chased into functional spliceosomes (*Seraphin and Rosbash, 1989*; *Séraphin et al., 1988*). However, both native gel analysis of CC formation and recent single molecule analysis of U1 binding dynamics showed that many U1/pre-

*For correspondence:
ahoskins@wisc.edu

Competing interests: The authors declare that no competing interests exist.

**eLife digest** Our genes contain coded instructions for making the molecules in our bodies, but this information must be extensively processed before it can be used. The instructions from each gene are first copied into a molecule called a pre-mRNA, before a process known as splicing removes certain sections to form a mature mRNA molecule. Splicing can remove different sections of the pre-mRNA to make different mRNA molecules from the same gene depending on the current needs of the cell.

Splicing is controlled by a combination of proteins and other molecules, collectively called the spliceosome. A part of the spliceosome called U1 recognizes the start of pre-mRNA sections that need to be removed, which is referred to as the five-prime splice site (or "5' SS" for short). The attachment of U1 to such a site allows other molecules to also attach to the pre-mRNA, which eventually assemble a spliceosome. The very first steps in this process involve U1 and a set of other proteins that create what is called the "Early" or "E" complex. Although there are many molecules involved in the E complex, it was not known how they interact with each other and how this affects which splice sites are used for splicing in different cells.

Using advanced microscopy, Larson and Hoskins examined individual U1 molecules from yeast cells while the molecules formed E complexes and identified two different ways U1 can bind to five-prime splice sites. One process involved U1 attaching to pre-mRNA for a short time, whilst the other involved a longer association between U1 and pre-mRNA. Sometimes U1 could also transition between the first process and the second. The results showed that other parts of the E complex affected which process was used at different sites by affecting the type or duration of U1's attachment.

All U1 particles use the same components to attach to splice sites in all pre-mRNAs, but the most used splice sites are not always those that are predicted to have the strongest attachments to U1. This work helps to reveal how other proteins involved in splicing influence this effect, altering U1's ability to attach to pre-mRNAs to suit each new situation. This also allows cells to change gene splicing to fit different situations. Many genes in our bodies rely on splicing and understanding this process in detail could be the key to diagnosing and treating a range of different illnesses.
DOI: https://doi.org/10.7554/eLife.27592.002

mRNA interactions are short-lived (*Ruby, 1997*; *Hoskins et al., 2011*). Since these experiments did not assay ECPs, it is unknown if these transient U1 binding events originated from the lack of simultaneous binding of both U1 and ECPs or if E complex itself is dynamic.

Within E complex, U1 recognizes the 5' SS by base pairing between the 5' end of the U1 snRNA and the transcript (*Mount et al., 1983*; *Siliciano and Guthrie, 1988*; *Zhuang and Weiner, 1986*). Despite the high conservation of this region of the U1 snRNA, 5' SS can vary greatly in sequence. In yeast, most pre-mRNAs contain strong 5' SS that match the consensus, 5'-GUAUGU-3' (*Grate and Ares, 2002*). However, there are several non-consensus 5' SS that are also found in yeast introns and are predicted to make fewer base pairing interactions with U1 (*Grate and Ares, 2002*; *Qin et al., 2016*). These introns may be less efficiently spliced and often function as 'weak' 5' SS signals for U1. In humans, 5' SS are often weak and >9000 sequence variants have been reported (*Roca and Krainer, 2009*). How U1 uses a single snRNA to facilitate spliceosome assembly at many different 5' SS of varying strengths is not yet clear.

Genetic evidence from yeast indicates that ECPs may aid U1 in 5' SS recognition. ECPs are essential for the splicing of particular transcripts (*Hossain et al., 2009*; *Qin et al., 2016*) or in the presence of certain U1 protein or snRNA mutations (*Schwer and Shuman, 2014*; *Chang et al., 2010*; *Liao et al., 1993*; *Schwer et al., 2013*). The genetic data indicate an apparent redundancy of ECPs in their ability to compensate for U1 mutations and loss of multiple interactions is often necessary to impair splicing (*Chang et al., 2012*; *Schwer and Shuman, 2014*; *Schwer et al., 2013*). Many genetic interactions between U1 and ECPs converge on both the 5' end of the snRNA and the Yhc1 protein (U1-C in humans) (*Schwer and Shuman, 2014*; *Schwer et al., 2013*). Crystal structures of the human U1 snRNP have revealed that U1-C abuts a portion of the snRNA/5' SS duplex, and biochemical data show that U1-C can modulate the stability of duplexes formed with weak 5' SS (*Kondo et al.,*

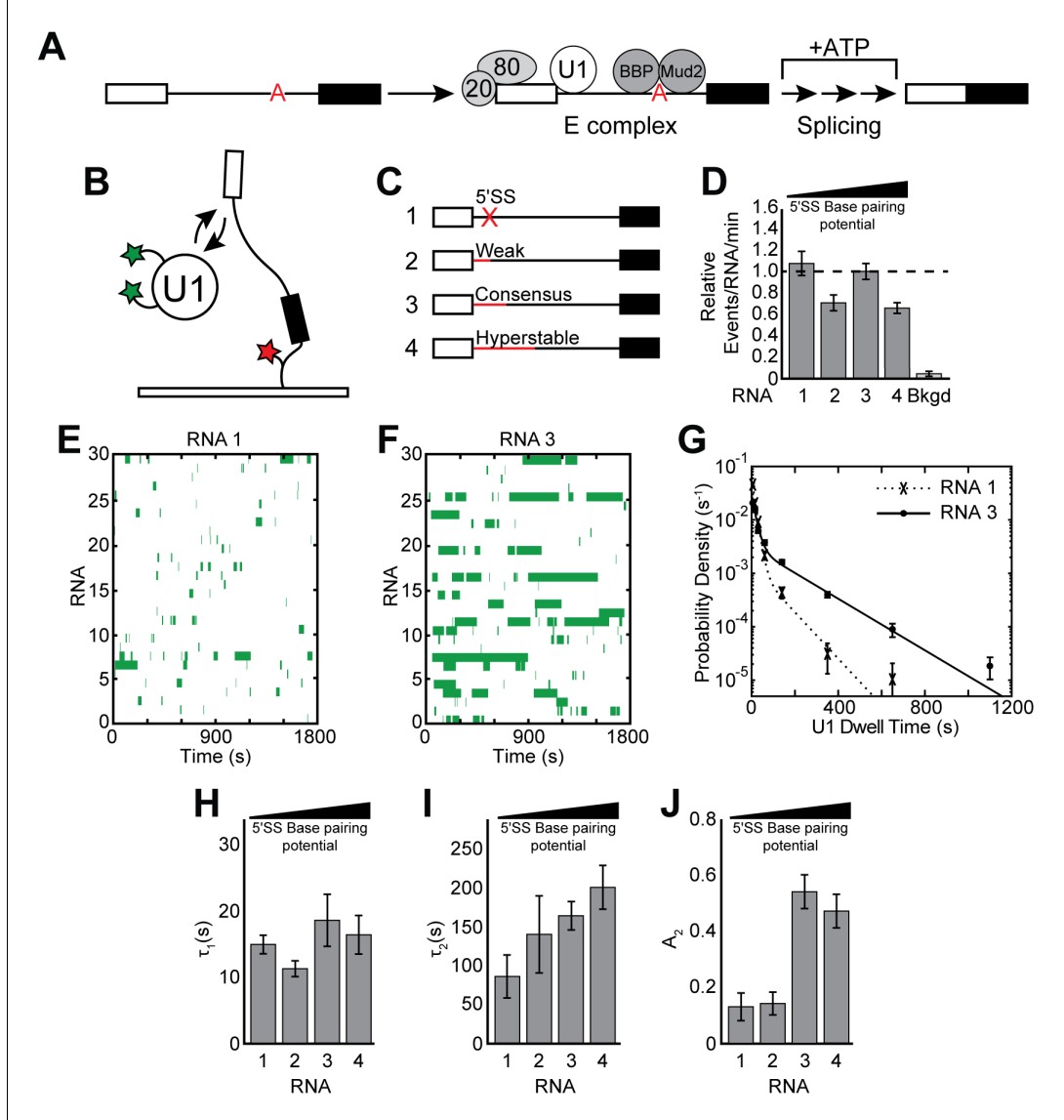

**Figure 1.** Cartoon of E complex formation and the impact of the 5' SS mutations on U1 binding kinetics. (A) E complex assembles on pre-mRNA and contains U1 and BBP/Mud2 associated with the 5' SS and BS, respectively. In yeast, the nuclear CBC is composed of Cbp20 and Cbp80, associates with the pre-mRNA 5' cap, and is also a member of E complex. Subsequent steps then lead to splicing. (B) Cartoon of a two-color CoSMoS experiment for observing U1 binding dynamics. The U1 snRNP contained two green-excited fluorophores attached to two different proteins while the RNA was immobilized to the slide surface and contained a single Cy5 fluorophore. (C) Graphic representation of capped RNAs with variable 5' SS used in these experiments and their corresponding label number. RNA sequences are given in *Supplementary file 1*. (D) Bar graph comparing the relative number of U1 binding events observed on each RNA depicted in panel (C). (E) Rastergram depicting U1 binding events on RNAs lacking a functional 5' SS (RNA 1). The rastergram represents a stack of 30 time ribbons from individual RNA molecules in which U1 binding events are shown as green bands. (F) Rastergram depicting U1 binding events on RNAs containing a functional, consensus 5' SS (RNA 3). (G) Probability density histogram of dwell times for U1 on RNAs lacking (RNA 1) and containing (RNA 3) a functional 5' SS. Lines represent fits of the distributions of dwell times to equations containing two exponential terms. (H–J) Bar graph comparison of the fit parameters ($\tau_1$, panel H; $\tau_2$, panel I; the $\tau_2$ amplitude $A_2$, panel J) obtained from analysis of the dwell time distributions of U1 binding events on RNAs 1–4. Details of the fit parameters for data shown in (G–J) can be found in *Supplementary file 2*. Error bars in (D, G) represent the error in counting statistics as given by the variance of a binomial distribution. Bars in (H–J) represent the fit parameters ± S.D.

DOI: https://doi.org/10.7554/eLife.27592.003

The following figure supplements are available for figure 1:

**Figure supplement 1.** Examples of fluorescence intensity traces supplementing data shown in *Figure 1C–J* showing individual U1-SNAP$_f$ subcomplexes co-localizing with the indicated surface-tethered RNAs containing variable 5' SS sequences.

DOI: https://doi.org/10.7554/eLife.27592.004

*Figure 1 continued*

**Figure supplement 2.** RNAs containing a weak 5′ SS splice in vitro.
DOI: https://doi.org/10.7554/eLife.27592.005

**Figure supplement 3.** Single molecule analysis of U1 binding after ablation of the 5′ end of the snRNA.
DOI: https://doi.org/10.7554/eLife.27592.006

*2015*; *Pomeranz Krummel et al., 2009*). Moreover, mutations in yeast Yhc1 can either bypass the requirement for the Prp28 ATPase in U1/U6 exchange during activation or exacerbate the phenotypes of defective Prp28 mutants (*Schwer and Shuman, 2014*; *Chen et al., 2001*). Recently, Prp28 has also been shown to play an ATP-independent role in E complex formation (*Price et al., 2014*). Together these data imply that the Yhc1/U1-C protein plays a critical role in U1's interactions with 5′ SS and ECPs may modulate this function. Despite these genetic and biochemical data, key mechanistic questions of how U1 interacts with RNA remain unanswered. For example, it is not clear how snRNP proteins facilitate base pairing between the snRNA and intron or how non-U1 splicing factors strengthen or weaken U1 binding to regulate 5′ SS selection.

To understand the steps involved in 5′ SS recognition, it is essential to characterize the interaction kinetics between U1 and pre-mRNAs and how these dynamics are influenced by both *cis*- and *trans*-acting factors. We have used colocalization single molecule spectroscopy (CoSMoS) to observe U1 interactions with RNAs containing or lacking a consensus 5′ SS as well as to monitor E complex formation between fluorescently-labeled U1, BBP, and pre-mRNAs in yeast whole cell extract (WCE). Our data reveal how dynamic E complex formation tunes U1 interactions at the 5′ SS and define a pathway for efficient recruitment and retention of U1 on intron-containing transcripts.

## Results

### 5′ SS pairing is critical for long-lived U1 binding events

To determine how U1 binding kinetics are influenced by splice site strength, we carried out CoSMoS experiments to measure the dwell times of individual, fluorescent U1 molecules on surface-immobilized RNA substrates in WCE (*Figure 1B*). In these experiments, endogenous U1 snRNPs were fluorescently labeled on two protein components harboring C-terminal $SNAP_f$ tags (*Hoskins et al., 2011*). U1 binding was observed by colocalization of fluorescence from U1 molecules with surface immobilized RNAs (*Hoskins et al., 2011*). We used the well-characterized, model pre-mRNA substrate RP51A (*Pikielny and Rosbash, 1986*) in these experiments and varied its consensus 5′ SS sequence by reducing or increasing the number of potential base pairs with the U1 snRNA. The resulting set of RNAs (*Figure 1C* and *Supplementary file 1*) were 5′ capped and identical in sequence with the exception of the 5′ SS which was either non-functional (RNA 1), weak (RNA 2), the yeast consensus sequence (RNA 3), or hyperstabilized to be completely complementary to the 5′ end of the U1 snRNA (RNA 4).

It has previously been shown that U1 associates with immobilized RNAs containing a 5′ SS and that removal of a functional 5′ SS results in decreased surface accumulation (*Hoskins et al., 2011*). However, the lifetimes of U1 interactions on these different RNAs were not compared with one another. We carried out this kinetic analysis under conditions in which the WCE was depleted of ATP by addition of glucose, which permits U1 binding and E complex formation but not subsequent steps in spliceosome assembly (*Seraphin and Rosbash, 1989*). We observed frequent, transient interactions between U1 and RNAs either containing or lacking a functional 5′ SS (*Figure 1D*, *Figure 1—figure supplement 1*). We compared these interactions using rastergrams that simultaneously depict U1 binding events occurring on multiple RNAs. In agreement with previous results, it was apparent that U1 interacted only briefly with RNAs that lacked a 5′ SS (*Figure 1E*) but stably associated with RNAs containing a consensus 5′ SS (*Figure 1F*).

On both RNAs the distribution of observed dwell times was best fit using a double exponential function containing short ($\tau_1$ with amplitude $A_1$) and long ($\tau_2$ with amplitude $A_2$) kinetic parameters (*Figure 1G–J* and *Supplementary file 2*). As expected from the rastergrams, removal of the functional 5′ SS resulted in a large decrease in both $\tau_2$ and its amplitude, $A_2$, and an accompanying increase in $A_1$ (*Figure 1I,J* and *Supplementary file 2* ). However, we were surprised to observe that

the overall number of binding events showed only a modest decrease on RNAs lacking a 5′ SS (*Figure 1D*), and little change was observed in the short kinetic parameter ($\tau_1$) (*Figure 1H*). Background binding resulting from fleeting association of fluorescent U1 molecules with the slide surface was rarely seen and could not account for the number of events that were observed (*Figure 1D*). U1 frequently colocalizes with immobilized RNAs even in the absence of a functional 5′ SS.

We wondered if use of a weak 5′ SS would result in U1 binding kinetics more similar to those observed with the consensus sequence or those seen when the RNA lacked a 5′ SS altogether. We tested this using RNA 2 which contains the 5′ SS normally found in the yeast SUS1 pre-mRNA (*Figure 1C*). This 5′ SS can form fewer potential base pairs with U1 relative to the consensus (4 vs. 6 base pairs, respectively) but permits splicing in vitro when placed into RP51A (*Figure 1—figure supplement 2*). The results were similar to those observed in the absence of a functional 5′ SS with short-lived interactions occurring frequently (*Figure 1D*) and predominating the distribution. Long-lived events were rare, resulting in an $A_2$ value nearly identical to that observed in the absence of a 5′ SS and much lower than observed with the consensus (*Figure 1J*). Ablation of the 5′ end of the U1 snRNA with RNase H also resulted in nearly complete loss of the long-lived binding events and frequent short-lived interactions (*Figure 1—figure supplement 3*). These data suggest that both binding lifetime as well as the relative abundances of short- and long-lived binding events are influenced by 5′ SS strength and the ability to pair with the snRNA.

If eliminating or weakening the 5′ SS reduces the longest-lived U1 events, we predicted that strengthening the 5′ SS should do the opposite. We analyzed U1 interactions with RNAs containing a hyperstabilized 5′ SS, capable of making up to 10 potential base pairs with the snRNA (*Figure 1C*, RNA 4). In this case, we frequently observed both short- and long-lived interactions (*Figure 1—figure supplement 1D*). Unexpectedly, the parameters obtained from a fit of the dwell time distribution revealed only a slight increase in $\tau_2$ while its amplitude ($A_2$) and $\tau_1$ remained close to what were observed with the consensus 5′ SS (*Figure 1H–J*). Long-lived U1 binding events seem to be much more sensitive to a loss in base pairing than an increase, while short-lived events are not solely dependent on the ability to base pair with a strong 5′ SS.

## The RNA cap and BS facilitate U1 recruitment in the absence of a strong 5′ SS

Do the short-lived U1 binding events represent specific or non-specific interactions between the snRNP and RNA? It has previously been proposed that ECPs facilitate U1 recognition of splice sites, particularly those with limited complementarity to the snRNA (*Qiu et al., 2012*; *Hossain et al., 2009*), as well as splicing-independent occupancy of U1 on transcripts (*Görnemann et al., 2005*; *Patel et al., 2007*; *Spiluttini et al., 2010*). It is possible that momentary U1 binding results from interactions between U1 and ECPs bound to the RNA.

We tested the role of ECPs in U1 recruitment by mitigating the influence of both the CBC and BBP. Previous work has shown that the impact of the CBC on splicing in vitro can be inhibited by addition of cap analog dinucleotide (CA), which acts as a competitive inhibitor of capped RNAs for binding the CBC (*Edery and Sonenberg, 1985*; *Konarska et al., 1984*; *Lin et al., 1985*). Therefore, we carried out single molecule experiments either in the absence or presence of CA in order to remove the influence of the CBC (*Figure 2A*).

We hypothesized that the influence of BBP could be alleviated by mutation of the BS sequence in the transcript to prevent stable BBP association. We tested this using a CoSMoS assay for BBP binding to immobilized RNAs (*Figure 2—figure supplement 1*). On WT RP51A (RNA 3), we observed both short- and long-lived binding of SNAP$_f$-tagged BBP (*Figure 2—figure supplement 1D* and *Supplementary file 3*). Mutation of the BS eliminated the long-lived binding events but short-lived binding remained unchanged (*Figure 2—figure supplement 1F*; RNA 10). These short-lived events could be eliminated by simultaneous mutation of both the BS and a nearby $\Psi$BS (*Séraphin and Rosbash, 1991*; *Figure 2—figure supplement 1C*; RNA 7). This suggests that a 5′ SS and interactions with U1 alone cannot result in detectable BBP binding, in contrast with a previous model obtained by cross-linking in vivo that described U1-facilitated BBP recruitment (*Görnemann et al., 2005*). It is possible that irreversible cross-linking may have captured interactions too transient for us to observe or interactions occurring between BBP and RNA sequences other than the BS (i.e., $\Psi$BS). Regardless, since BBP appears to associate stably with the BS and briefly with the $\Psi$BS, we were able to mitigate its influence on U1 by mutation of both of these sequences.

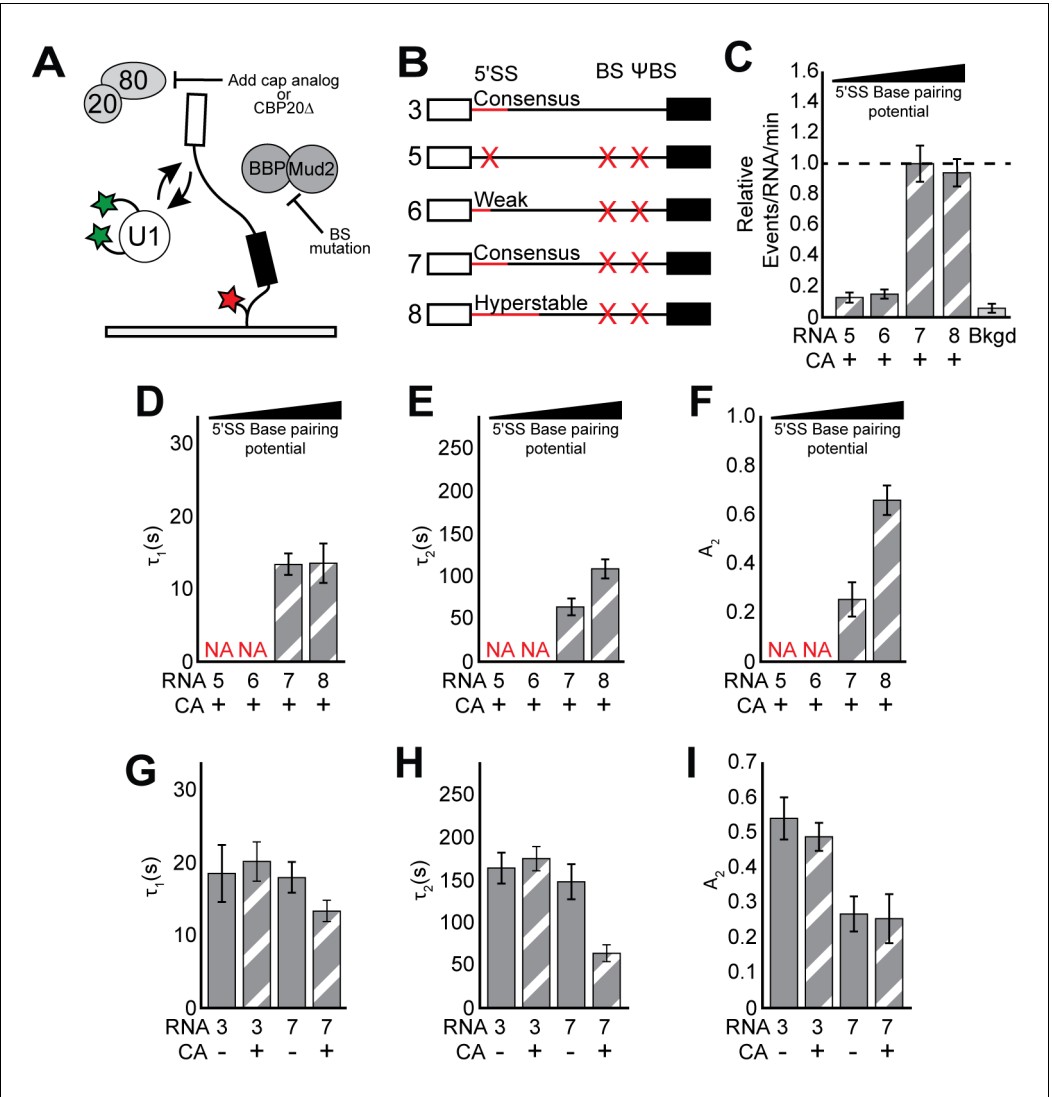

**Figure 2.** CoSMoS experiments monitoring U1 binding without the influence of ECPs. (**A**) Cartoon of a two-color CoSMoS experiment for observing U1 binding dynamics while mitigating the influence of ECPs. Addition of cap analog (CA) or genetic deletion of Cbp20 prevents the CBC from interacting with the immobilized RNAs, and deletion of the BS and ΨBS prevent stable binding of BBP/Mud2. (**B**) Graphic representation of capped RNAs with variable 5′ SS and BS and their corresponding label number. RNA sequences are given in *Supplementary file 1*. (**C**) Bar graph comparing the relative number of U1 binding events observed on RNAs 5–8 depicted in panel (**B**) and in the presence of CA. (**D–F**) Bar graph comparison of the fit parameters ($\tau_1$, panel D; $\tau_2$, panel E; the $\tau_2$ amplitude $A_2$, panel F) obtained from analysis of the dwell time distributions of U1 binding events on RNAs 5–8 in the presence of CA. (**G–I**) Bar graph comparison of the fit parameters ($\tau_1$, panel G; $\tau_2$, panel H; the $\tau_2$ amplitude $A_2$, panel I) obtained from analysis of the dwell time distributions of U1 binding events on RNAs 3 or 7 in the presence or absence of CA. Details of the fit parameters for data shown in (**C–I**) can be found in *Supplementary file 2*. Error bars in (**C**) represent the error in counting statistics as given by the variance of a binomial distribution. Bars in (**D–I**) represent the fit parameters ± S.D. Striped bars in (**C–I**) indicate the addition of CA in those experiments.

DOI: https://doi.org/10.7554/eLife.27592.007

The following figure supplements are available for figure 2:

**Figure supplement 1.** Single molecule analysis of BBP-SNAP$_f$ binding dynamics.
DOI: https://doi.org/10.7554/eLife.27592.008

**Figure supplement 2.** Examples of fluorescence intensity traces supplementing data shown in *Figure 2C–I* showing individual U1-SNAP$_f$ subcomplexes co-localizing with the indicated surface-tethered RNAs containing variable 5′ SS sequences in the presence of CA and in the absence of the BS/ΨBS.

*Figure 2 continued on next page*

*Figure 2 continued*

DOI: https://doi.org/10.7554/eLife.27592.009
**Figure supplement 3.** Impact of CBP20 deletion on U1 binding dynamics and comparison with CA addition.
DOI: https://doi.org/10.7554/eLife.27592.010

In the presence of CA, colocalization of U1 with the tethered RNAs became strongly dependent on the 5' SS when those RNAs also lacked the BS/ΨBS sequences. We no longer observed colocalization of U1 with RNAs that either lacked a functional 5' SS or contained the weak 5' SS (*Figure 2B, C* and *Figure 2—figure supplement 2*; RNAs 5 and 6). In addition, U1 binding was near background levels after ablation of the 5' end of the snRNA and ECP mitigation, even when the RNA contained a consensus 5' SS (*Figure 1—figure supplement 3C*). Any remaining U1/RNA binding events occurred too quickly for us to detect in these experiments, likely with lifetimes <~0.5 s. These data suggest that the 5' cap and BS sequence are responsible for recruitment of U1 on RNAs that lack 5' SS (*Figure 1D*) or when the 5' SS pairing region of the snRNA has been removed. In contrast, frequent binding was still observed on RNAs containing strong 5' SS and with the intact snRNA (*Figure 2B*, RNAs 7 and 8, and *Figure 1—figure supplement 3*). Thus, short-lived binding events are not non-specific - they depend on features of the RNA and likely involve interactions between U1 and ECPs bound to the transcript. When a 5' SS is present, short-lived binding events originate from both interactions between U1 and the 5' SS and interactions between U1 and ECPs. In the absence of a 5' SS or intact snRNA, U1 is recruited to RNAs primarily through interactions with ECPs.

## The RNA cap and BS modulate U1 binding at strong splice sites

While a strong 5' SS is sufficient for U1 recruitment, binding of U1 was significantly weakened when ECPs were mitigated (*cf. Figure 2D–F* vs. *Figure 1H–J* and *Supplementary file 2*). On RNAs containing a consensus 5' SS, the long-lived parameter $\tau_2$ was reduced ~3 fold and these long-lived events were much less frequent. In fact, U1 dwell times on RNAs containing a 5' SS but without ECPs were similar to those observed on RNAs lacking a 5' SS but with ECPs. This indicates that U1/ECP interactions play an important role in tuning U1 binding even at strong 5' SS and that the longest-lived U1 complexes depend on both the 5' SS and ECPs.

After ECPs were mitigated, the differences between RNAs containing a consensus or hyperstabilized 5' SS became apparent. Hyperstabilization did not change $\tau_1$ (*Figure 2D*). However, both $\tau_2$ and $A_2$ increased approximately 2-fold upon hyperstabilization relative to the consensus, a greater change than was observed when the ECPs were not mitigated (*cf. Figure 1I and J*, RNA 3 and 4 with *Figure 2E and F*, RNA 7 and 8). These data show that increased pairing potential between U1 and the 5' SS predominantly impacts the kinetics of the long-lived binding events and ECPs mask this effect.

## The BS and cap modulate U1 binding by distinct mechanisms

We next tested if the cap and BS/ΨBS independently influence U1 binding dynamics. Unexpectedly, while the lifetimes of the longest-lived U1 binding events ($\tau_2$) could be modulated by either the cap or the BS/ΨBS, this was not true of their relative abundance ($A_2$). In the presence of CA, we observed an increase in both $\tau_2$ and $A_2$ for U1 binding events on RNAs containing a BS/ΨBS compared to those that lacked these sequences (*Figure 2H,I*; RNAs 3 and 7 + CA). In contrast, addition of CA changed $\tau_2$ but not $A_2$ on RNAs that lacked the BS/ΨBS (*Figure 2H,I*; RNA 7 and ±CA). Similar results were also obtained using WCE prepared from strains in which the small subunit of the CBC was genetically deleted (*Figure 2—figure supplement 3* and *Supplementary file 4*). Thus, addition of CA only decreases U1 lifetimes on RNAs lacking a BS/ΨBS and removal of the BS/ΨBS only decreases U1 lifetimes in the presence of CA. However, the BS/ΨBS additionally changes $A_2$ independent of the presence or absence of CA.

## BBP associates with the longest-lived U1/Pre-mRNA complexes

The above results show that molecular features of the RNA other than the 5' SS can influence U1 binding. It is likely this occurs by interactions between U1 and ECPs— protein-protein contacts with CBC associated at the 5' cap and BBP/Mud2 associated at the BS. We therefore tested whether or

not the longest-lived U1 events were correlated with simultaneous binding of the transcript by ECPs. CBC is very abundant in WCE and has so far proven intractable for CoSMoS studies (data not shown). However, BBP is amenable to C-terminal SNAP$_f$ tagging and binds surface-immobilized RNAs with sequence specificity (*Figure 2—figure supplement 1*). We engineered a yeast strain containing two DHFR-tags on U1 and a SNAP$_f$ tag on BBP to permit simultaneous visualization of U1 and BBP (*Supplementary file 4*). WCE prepared from this strain formed CC and retained splicing activity in vitro (*Figure 3—figure supplement 1* and *Figure 3—figure supplement 2*). We then carried out 3-color CoSMoS assays to monitor U1 and BBP binding dynamics on immobilized pre-mRNAs that contained a consensus 5′ SS and only the BS (*Figure 3A*).

U1 and BBP frequently colocalized to the same pre-mRNA (*Figure 3B,C* and *Figure 3—figure supplement 3*). Not all U1 or BBP binding events resulted in colocalization of the other component: U1 often bound without BBP and *vice versa*. However, colocalized complexes of U1, BBP, and pre-mRNA could persist from a few seconds to several minutes (*Figure 3C*). Since long-lived U1 and BBP complexes are dependent on the presence of a 5′ SS (*Figure 1*) or BS (*Figure 2—figure supplement 1*), respectively, it is likely that many of these co-complexes of U1, BBP, and pre-mRNA represent the spliceosome E complex with U1 engaged with the 5′ SS and BBP bound at the BS.

To test whether or not the longest-lived U1 binding events are correlated with BBP occupancy, we measured the dwell times of U1 molecules colocalizing with BBP and compared those to U1 dwell times that showed no evidence for coincident BBP binding (*Figure 3D*). The mean lifetime of U1 molecules colocalizing with BBP was significantly longer (182.3 ± 35.6 s) compared to U1 molecules binding in the absence of BBP (73.6 ± 32.2 s), while no difference was seen in a randomized control (*Figure 3—figure supplement 4*). This supports the notion that simultaneous binding of the transcript by BBP promotes a functional change in U1 that promotes stable association and long-lived binding.

In these three-color experiments, CBC could still potentially interact with the RNA and U1 independent of BBP. When BBP was not simultaneously present with U1 on the RNA, the influence of the CBC alone likely resulted in the ~74 s mean U1 lifetime. To confirm this, we compared this value to calculated mean U1 lifetimes for interactions on RNAs under conditions in which only the CBC was allowed to bind (RNA 7; 82.6 ± 11.0 s) or when binding of both BBP and CBC was prevented (RNA 7, +CA; 27.4 ± 2.6 s) using two-color experiments and data described in *Figure 2*. Only under the former condition is the average U1 lifetime similar to what we observed in the three-color experiments in the absence of simultaneous BBP association. Thus, U1 lifetimes when BBP binding is either not observed on the RNA (three-color experiments) or not permitted due to a BS mutation (two-color experiments) are likely due to the influence of the CBC and are much longer than U1 binding in the absence of both the CBC and BBP.

## E complex forms transiently to enable 5′ SS and BS redefinition

These CoSMoS experiments also revealed the lifetime and dynamics of the spliceosome E complex. Consistent with the multi-exponential kinetics observed with both U1 (*Figure 1G*) and BBP (*Figure 2—figure supplement 1G*), the distribution of E complex lifetimes was also best described by a function containing two exponential terms ($\tau_1$ of ~20 s and $\tau_2$ of ~165 s, *Figure 3E*, *Figure 3—figure supplement 5*, and *Supplementary file 5*). This indicates that multiple types of E complexes exist that contain both U1 and BBP. The longest-lived complexes that contribute to $\tau_2$ may be the same as those that result in commitment and are observable by native PAGE.

We next analyzed the pathways of E complex assembly and disassembly. E complex assembly can proceed by either initial U1 or BBP binding (*Figure 3—figure supplement 6*). After E complex formation, we observed frequent redefinition of the 5′ SS and BS as U1 or BBP spontaneously dissociated and another molecule subsequently bound (*Figure 3C*). For example, we noted the formation of E complexes that would persist for several minutes before loss of U1 followed by rebinding of a different U1 molecule. This resulted in redefinition of the 5′ SS while definition of the BS was maintained by BBP. The alternative pathway of BS redefinition was also observed as was switching between 5′ SS and BS redefinition on the same pre-mRNA molecule. In both cases, analysis of fluorescence trajectories supported recruitment of only a single U1 molecule and a single BBP molecule to the vast majority of pre-mRNAs at any one time (*Figure 3—figure supplement 7*). This signifies that redefinition requires prior release of either U1 or BBP from the 5′ SS or the BS, respectively.

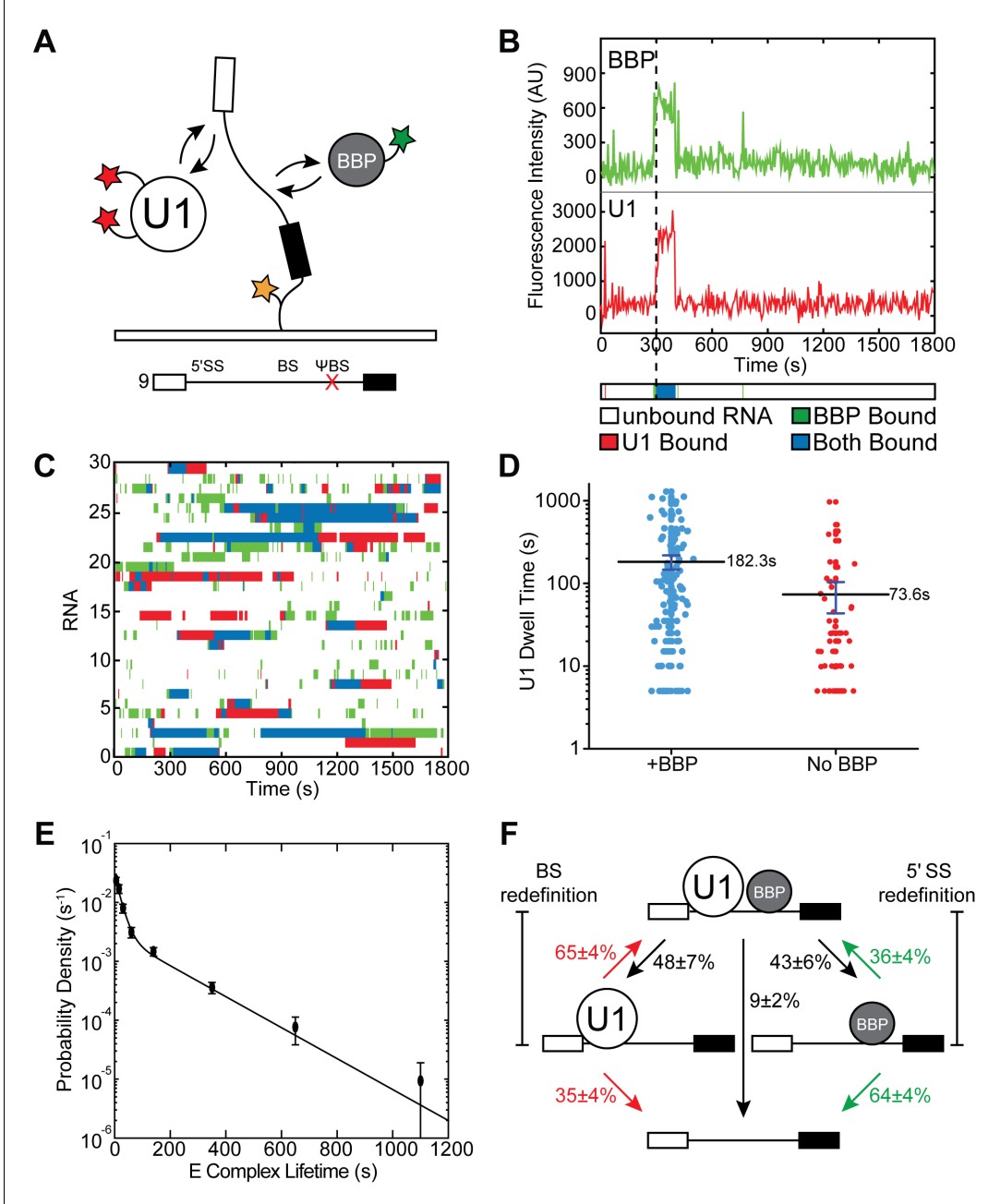

**Figure 3.** Three-color CoSMoS observation of U1 and BBP binding dynamics. (**A**) Schematic of a three-color experiment in which U1 was labeled with two red-excited (Cy5) fluorophores, BBP was labeled with a single green-excited (Dy549) fluorophore, and the surface-tethered pre-mRNA was labeled with an Alexa488 fluorophore. Shown below is a graphic representation of the capped, immobilized RNA used in these three-color experiments. (**B**) Representative time records showing peaks in fluorescence intensity corresponding to colocalization of BBP (green) and U1 (red) with the same individual pre-mRNA molecule. The dashed line demarks the arrival of U1. Shown below the time record is the corresponding time ribbon in which U1 and BBP binding are represented by red and green bands, respectively. Colocalizations of U1 and BBP are represented by blue bands. (**C**) Rastergram depicting typical U1 and BBP binding events and colocalization on immobilized RNAs. (**D**) Measured U1 lifetimes in the presence or absence of BBP colocalization. The mean lifetimes are noted ±S.E. (**E**) Probability density histogram of dwell times for colocalized U1 and BBP complexes. The lines represent the fit of the distribution to an equation containing two exponential terms. Details of the fit parameters can be found in *Supplementary file 5*. (**F**) Routes for loss of either U1 or BBP fluorescent spots from immobilized RNAs following their colocalization. Percentages represent the fraction of U1/BBP complexes in which fluorescence disappeared by the indicated pathway ($N$ = 173 initial colocalized U1/BBP complexes). Error bars in (**E and F**) represent the error in counting statistics as given by the variance of a binomial distribution.

DOI: https://doi.org/10.7554/eLife.27592.011

*Figure 3 continued on next page*

*Figure 3 continued*

The following figure supplements are available for figure 3:

**Figure supplement 1.** Characterization of U1- and BBP-labeled yeast splicing extracts.
DOI: https://doi.org/10.7554/eLife.27592.012

**Figure supplement 2.** U1- and BBP-labeled WCE forms commitment complexes.
DOI: https://doi.org/10.7554/eLife.27592.013

**Figure supplement 3.** Examples of fluorescence intensity traces supplementing data shown in *Figure 3B* showing individual U1-DHFR subcomplexes and BBP-SNAP$_f$ co-localizing with surface-tethered RNAs.
DOI: https://doi.org/10.7554/eLife.27592.014

**Figure supplement 4.** Randomized control for analysis of U1 lifetimes during colocalization with BBP to complement *Figure 3D*.
DOI: https://doi.org/10.7554/eLife.27592.015

**Figure supplement 5.** E complex lifetimes on RNAs containing or lacking the ΨBS.
DOI: https://doi.org/10.7554/eLife.27592.016

**Figure supplement 6.** Pathways for E complex formation.
DOI: https://doi.org/10.7554/eLife.27592.017

**Figure supplement 7.** Counting statistics for the number of steps observed during loss of fluorescence from either U1-double DHFR or BBP-SNAP$_f$ binding events in three-color CoSMoS experiments.
DOI: https://doi.org/10.7554/eLife.27592.018

Since ATP was not included in these experiments, these redefinition events are unlikely to require ATPase activity.

Analysis of the fates of individual E complexes revealed that the loss of either U1 or BBP was nearly equally probable (*Figure 3F*). However, pre-mRNAs that lost U1 were twice as likely to also lose BBP rather than redefine the 5′ SS. In contrast, pre-mRNAs that lost BBP were nearly twice as likely to redefine the BS as they were to subsequently lose U1. This suggests that in vitro E complex formation results in different degrees of obligation to the 5′ SS and BS. In sum, these experiments reveal that E complex formation is dynamic and results in little commitment to any particular 5′ SS/BS pair.

## Yhc1 is essential for U1 recruitment to the 5′ SS

Our data support distinct short- and long-lived U1 interactions influenced by both 5′ SS sequence as well as ECPs. Structures of the human U1 snRNP place the U1-C protein (yeast Yhc1) at the site of duplex formation between the snRNA and 5′ SS (*Kondo et al., 2015*), and there is strong genetic and biochemical evidence to support links between Yhc1, ECPs, and stability of the snRNA/5′ SS duplex (*Schwer and Shuman, 2014*; *Abovich and Rosbash, 1997*; *Hage et al., 2009*; *Schwer and Shuman, 2015*; *Zhang and Rosbash, 1999*). Therefore, it is possible that Yhc1 also plays a critical role in modulating U1 binding kinetics. We tested this by engineering a yeast strain that permits fluorescent labeling of the U1 snRNP with SNAP$_f$ tags as well as exchange of Yhc1 alleles by plasmid shuffling (*Supplementary files 4 and 6*). This allowed us to prepare WCEs containing fluorescent U1 in which every U1 molecule contained a Yhc1 protein harboring specific amino acid mutations near the site of snRNA/5′ SS duplex formation.

We introduced two well-characterized alleles of Yhc1 that have minimal impact on yeast viability but are predicted to have opposing impacts on U1 stability during splicing (*Figure 4A*). Yhc1-L13F permits bypass of Prp28 function during spliceosome activation, presumably due to destabilization of the U1 snRNA/5′ SS duplex at this stage (*Chen et al., 2001*). In contrast, Yhc1-D36A reinforces the need for Prp28, possibly due to stabilization of this same duplex (*Schwer and Shuman, 2014*). As expected, yeast containing SNAP$_f$-tagged U1 and these Yhc1 alleles were viable, and WCE prepared from these strains maintained high levels of splicing in vitro (*Figure 3—figure supplement 1*).

We first measured binding dynamics of the mutant U1 molecules on RNAs containing a consensus 5′ SS and BS/ΨBS in the absence of CA (*Figure 4B*, RNA 3 –CA). U1 binding dynamics were minimally perturbed by inclusion of the Yhc1-L13F mutation (*Figure 4C–F* and *Figure 4—figure supplement 1*). We observed only a slight reduction of $\tau_2$ and $A_2$ with Yhc1-L13F even though this mutation eliminates the ATP-dependence of U1 release during spliceosome activation. On the other hand, Yhc1-D36A showed little change in the long-lived parameters and instead more than doubled the short-lived parameter, $\tau_1$ (*Figure 4D*).

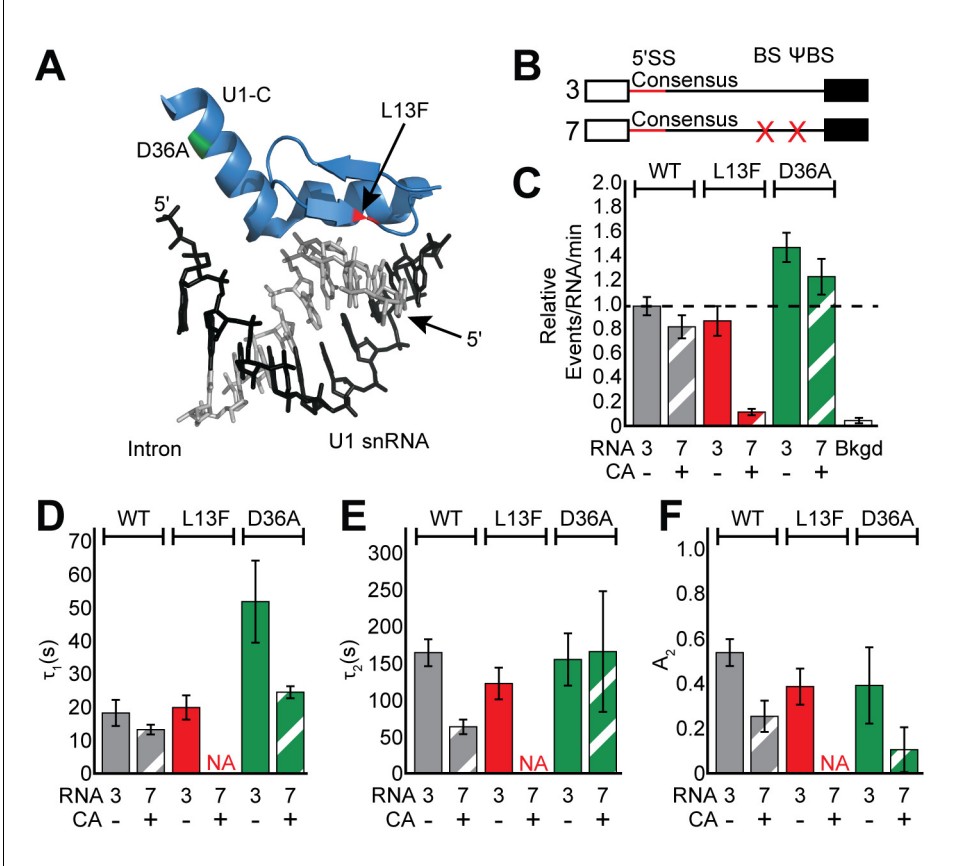

**Figure 4.** Impact of Yhc1 mutations on U1 binding dynamics. (**A**) Close-up view of the structure of a human U1 snRNP fragment focusing on proximity of the U1 snRNA/5′ SS duplex to the U1-C protein (from 4PJO.pdb). The locations of the L13F (red) and D36A (green) mutations incorporated into yeast Yhc1 are shown. (**B**) Graphic representation of capped RNAs with consensus 5′ SS and variable BS and their corresponding label number. RNA sequences are given in *Supplementary file 1*. (**C**) Bar graph comparing the relative number of WT and Yhc1 mutant U1 binding events observed on RNAs 3 and 7 depicted in panel (**B**) and in the presence or absence of CA. (**D–F**) Bar graph comparison of the fit parameters ($\tau_1$, panel D; $\tau_2$, panel E; the $\tau_2$ amplitude $A_2$, panel F) obtained from analysis of the dwell time distributions of WT and Yhc1 mutant U1 binding events on RNAs 3 and 7 in the presence or absence of CA. Details of the fit parameters for data shown in (**D–F**) can be found in *Supplementary file 2*. Error bars in (**C**) represent the error in counting statistics as given by the variance of a binomial distribution. Bars in (**D–F**) represent the fit parameters ± S.D. Striped bars in (**C–F**) indicate the addition of CA in those experiments.
DOI: https://doi.org/10.7554/eLife.27592.019

The following figure supplement is available for figure 4:

**Figure supplement 1.** Examples of fluorescence intensity traces supplementing data shown in *Figure 4C–F* showing individual Yhc1-L13F or –D36A U1-SNAP$_f$ subcomplexes co-localizing with the indicated surface-tethered RNAs in the presence or absence of CA (RNAs 3 and 7, *Figure 4B*).
DOI: https://doi.org/10.7554/eLife.27592.020

When the BS/ΨBS were removed and CA added (*Figure 4B*, RNA 7 + CA), the impact of the Yhc1-L13F mutation became more striking. Under these conditions, we were unable to observe binding of U1 to RNAs even though they contained a strong, consensus 5′ SS (*Figure 4C* and *Figure 4—figure supplement 1*). These data agree with previous analysis of the L13F mutation using short RNA fragments (*Du et al., 2004*) and suggest that ECPs become essential for recruitment of U1 to pre-mRNAs when U1/5′ SS pairing is destabilized by the Yhc1-L13F mutation. U1 molecules containing Yhc1-D36A were still able to bind RNAs after the BS/ΨBS were removed and CA was added (*Figure 4C*). However, the increase in $\tau_1$ seen in the presence of ECPs was no longer apparent (*Figure 4D–F*). Together data from the Yhc1-L13F and -D36A mutations show that Yhc1 is likely

directly involved in both short- and long-lived U1 binding. Importantly, the impact of Yhc1 mutation on U1 binding can be modified by the presence or absence of ECPs.

## The RNA cap and BS facilitate U1 release when Yhc1 is mutated

Finally, we tested whether or not additional potential base pairing interactions between U1 and the 5′ SS could counteract the Yhc1-L13F mutation and enable cap- and BS/ΨBS-independent recruitment. Indeed, Yhc1-L13F U1 readily colocalized with RNAs containing a hyperstabilized 5′ SS even in the absence of the BS/ΨBS and in the presence of CA (*Figure 5A,B*). Unexpectedly, Yhc1-L13F U1 binding was much more stable than WT U1 under these conditions and persisted in some cases for tens of minutes (*Figure 5C,D* and *Figure 5—figure supplement 1*). Analysis of the dwell times revealed a dramatic increase in both $\tau_2$ and $A_2$ but only when the RNAs contained the hyperstabilized 5′ SS and the influence of ECPs was mitigated (*Figure 5E,F* and *Supplementary file 2*). These observations are consistent with previous reports of the L13F mutation facilitating additional base pairing interactions between U1 and the 5′ SS using a fragment of the RP51A pre-mRNA that lacked the BS sequence (*Du et al., 2004*).

These extremely long-lived Yhc1-L13F U1 binding events on hyperstabilized RNAs could only be completely suppressed by simultaneously permitting both BBP and the CBC to bind. Individually including the BS/ΨBS sequence or omitting CA did not result in a significant decrease in $\tau_2$ (*Figure 5—figure supplement 2*). Both conditions were required to reduce $\tau_2$ to values observed with WT U1; although, the BS/ΨBS alone influenced $A_2$. This is in agreement with our previous observation that hyperstabilization leads to an increase in WT U1 dwell time only after the influence of ECPs was mitigated (*Figure 1I,J* vs. *Figure 2E,F*). How ECPs mask the effects of hyperstabilization is not clear but could involve altered snRNP or RNA conformations that do not favor additional pairing interactions. While observations with Yhc1-L13F may be specific to this particular mutation, it illustrates how ECPs can tune U1 interactions in different ways depending on the composition of U1 and the 5′ SS to either enhance binding or promote release.

## Discussion

The importance of pairing between the U1 snRNA and 5′ SS in defining the intron as well as initiating spliceosome assembly has long been appreciated (*Lerner et al., 1980*; *Roca et al., 2013*). However, the extent of pairing by itself may not be informative of actual 5′ SS usage (*Roca et al., 2013*). Many consensus 5′ SS sequences are not used by the splicing machinery (*Berg et al., 2012*; *Roca et al., 2013*), and the 5′ SS is not predictive of U1 occupancy (*Görnemann et al., 2005*; *Patel et al., 2007*; *Spiluttini et al., 2010*; *Harlen et al., 2016*). Here, we have used single molecule fluorescence microscopy to demonstrate how splicing factors that are part of the spliceosome E complex (ECPs) modulate U1 binding at both consensus and non-consensus splice sites. U1 binding has been proposed to occur in multiple steps (*Du et al., 2004*; *McGrail et al., 2008*), and our single molecule kinetic data support U1 binding occurring with distinct intermediates. In our mechanism, initial but fleeting 5′ SS-dependent interactions transition into more stable complexes (*Figure 6A*). E complex tunes U1 occupancy on pre-mRNAs by altering the stability of complexes at each step as well as by regulating the shift between initial and stable binding.

### Properties of the U1/5′ SS Interaction

In the absence of ECPs, U1 recruitment is strictly 5′ SS-dependent, and the dwell times exhibit a biexponential distribution of short- and long-lived complexes (*Figure 2C–F*). Two simple kinetic schemes that could account for this observation are a two-step binding pathway (*Figure 6A*) or independent pathways that form two, non-interconverting U1/5′ SS complexes (*Figure 6—figure supplement 1*). While it is difficult to completely exclude the presence of independent pathways without knowledge of the forward and backward rates occurring at each step, we favor the two-step binding mechanism since it is supported by multiple aspects of our single molecule data. First, all U1/5′ SS interactions in the absence of ECPs are dependent on the 5′ SS sequence (*Figure 2C–F*), the Yhc1 protein (*Figure 4*), and the snRNA (*Figure 1—figure supplement 3*). This indicates that both short- and long-lived events are likely occurring at the site of snRNA/5′ SS duplex formation. A two-step binding mechanism does not require the formation of unrelated complexes at this critical location, as would the other model. Second, stabilization or destabilization of either the short- or long-lived

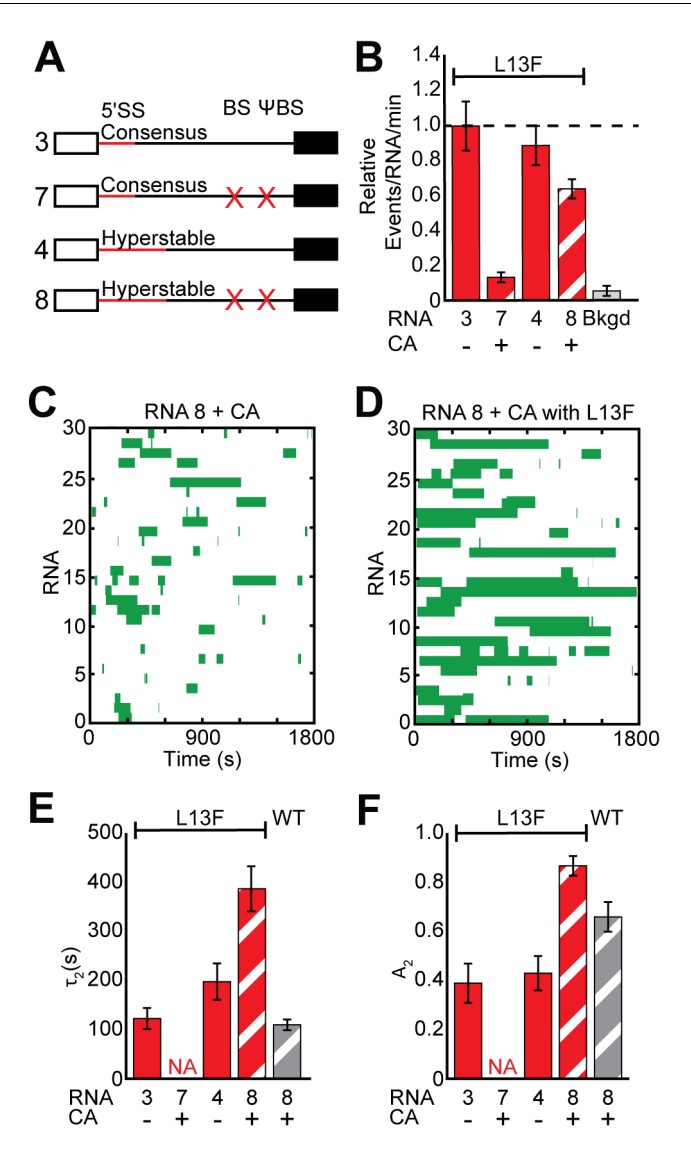

**Figure 5.** Impact of Yhc1 mutations on U1 binding dynamics to RNAs containing a hyperstabilized 5' SS. (A) Graphic representation of capped RNAs with variable 5' SS and BS and their corresponding label number. RNA sequences are given in *Supplementary file 1*. (B) Bar graph comparing the relative number of WT and Yhc1 mutant U1 binding events observed on RNAs depicted in panel (A) and in the presence or absence of CA. (C) Rastergram depicting WT U1 binding events on RNAs containing a hyperstabilized 5' SS (RNA 8) in the presence of CA. (D) Rastergram depicting Yhc1-L13F U1 binding events on RNAs containing a hyperstabilized 5' SS (RNA 8) in the presence of CA. (E, F) Bar graph comparison of the fit parameters ($\tau_2$, panel E; the $\tau_2$ amplitude $A_2$, panel F) obtained from analysis of the dwell time distributions of WT and Yhc1-L13F U1 binding events on RNAs shown in panel (A) in the presence or absence of CA. Details of the fit parameters for data shown in (E–F) can be found in *Supplementary file 2*. Error bars in (B) represent the error in counting statistics as given by the variance of a binomial distribution. Bars in (D,E) represent the fit parameters ± S.D. Striped bars in (B,E,F) indicate the addition of CA in those experiments.

DOI: https://doi.org/10.7554/eLife.27592.021

The following figure supplements are available for figure 5:

**Figure supplement 1.** Examples of fluorescence intensity traces supplementing data shown in *Figure 5B–F* showing individual Yhc1-L13F U1-SNAP$_f$ subcomplexes co-localizing with the indicated surface-tethered RNAs in the presence or absence of CA (RNAs 4 and 8, *Figure 5A*).

DOI: https://doi.org/10.7554/eLife.27592.022

*Figure 5 continued on next page*

*Figure 5 continued*

**Figure supplement 2.** The BS and 5' cap work toegether to promote Yhc1-L13F release from RNAs containing hyperstabilized 5' SS.

DOI: https://doi.org/10.7554/eLife.27592.023

complexes impacts both the complex lifetime as well as amplitude. For example, increasing the number of potential base pairs at the 5' SS in the absence of ECPs results in a simultaneous increase of more than 2-fold in both $\tau_2$ and $A_2$ without changing the binding event frequency (*Figure 2C–F*). This can be rationalized by two-step binding in which short-lived complexes are more likely to transition into long-lived complexes when additional pairing is possible, thus resulting in the decrease in $A_1$ and increase in $A_2$. Concerted changes in lifetimes and amplitudes are not as easily compatible with independent binding pathways (*Figure 6—figure supplement 1*) since the pathways are not kinetically coupled to one another. Finally, even under our most stabilizing conditions for U1 interaction, we still observe short-lived events (*Figure 5D*). This is expected if formation of the short-lived complex is a necessary precursor to more stable binding.

The molecular and functional differences between the initial and stable complexes are not clear. Results with the hyperstabilized 5' SS suggest that the transition between the two will involve a change in the snRNA/5' SS pairing region, as has been previously proposed (*Du et al., 2004*; *McGrail et al., 2008*). Our data do not preclude the possibility that short- or long-lived complexes may originate from altered interactions with other factors that transiently associate with U1 [for example, U2 (*Das et al., 2000*) or allow us to determine if different complexes promote subsequent steps in splicing to different degrees. Previous work showed that longer U1 lifetimes were correlated with intron loss (*Hoskins et al., 2011*). However, further experiments will be needed to determine the origin of those increased lifetimes and if they came about as a direct consequence of formation of the long-lived U1 complexes detected here. While the initial complex is short-lived, it is dependent on the presence of a strong 5' SS in the RNA. This property is distinct from the proposed U1 scanning complex (*McGrail et al., 2008*), which is predicted to involve snRNA nucleotides other

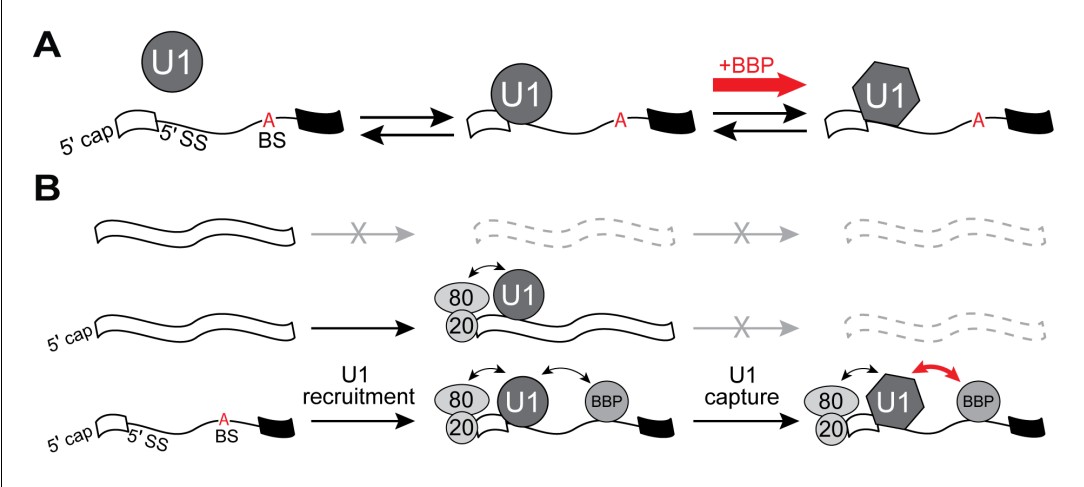

**Figure 6.** U1 binding kinetics are tuned to increase pre-mRNA competitiveness for the snRNP. (**A**) Model of a two-step mechanism for U1 binding. U1 first forms an initial complex that is short-lived before transitioning to a longer-lived complex. Binding of BBP to a downstream BS stimulates this transition. (**B**) A 'recruit and capture' model for stimulating U1 occupancy on pre-mRNAs. U1 is first recruited to transcripts by forming short-lived complexes dependent on interactions occurring either at potential 5' SS or by interactions with ECPs. Transition to long-lived complexes dependent snRNA/5' SS pairing and stimulated by BBP bound at the BS facilitates efficient capture of U1 by pre-mRNAs.

DOI: https://doi.org/10.7554/eLife.27592.024

The following figure supplement is available for figure 6:

**Figure supplement 1.** An alternative U1 binding mechanism to that shown in *Figure 6* that can give rise to multi-exponential dwell time distributions.

DOI: https://doi.org/10.7554/eLife.27592.025

than those that pair with the 5′ SS and may not recognize the 5′ SS itself. It is possible that scanning occurs prior to the 5′ SS dependent interactions that we observe.

## The spliceosome E complex is dynamic

Our single molecule data show that U1 still interacts dynamically with RNAs even while BBP is simultaneously bound (*Figure 3C*). This indicates that the spliceosome E complex is readily disassembled without ATP and only a subset of complexes are stable enough to result in commitment—in agreement with previous identification of the committed and uncommitted 'δ' complex by Ruby (*Ruby, 1997*). Moreover, our kinetic description of E complex lifetimes characterizes its heterogeneity and reveals the presence of both short- and long-lived E complexes (*Figure 3E*). We do not know if short-lived E complexes can transition into those that are long-lived. It is possible that E complexes of diverse stability originate from different levels of functional engagement of U1 and/or ECPs with the pre-mRNA or with one another.

One consequence of dynamic and reversible E complex formation is that it permits rapid definition and redefinition of the 5′ SS and BS that are subsequently used by the spliceosome. While BBP association with the pre-mRNA stimulates stable U1 binding (*Figure 6A*), the frequent release and rebinding of BBP while U1 remains bound suggests that U1 stabilization could be triggered by BS sequences not ultimately used by the spliceosome. Such a strategy could facilitate efficient spliceosome assembly by anchoring U1 at the 5′ SS while delaying BS definition until the optimal sequence for pairing with U2 is found. This flexibility may be important for regulating splicing, particularly on long introns which may contain many potential BS sequences. In the presence of ATP, it is likely that redefinition competes with spliceosome assembly at a particular 5′ SS/BS pair. Thus, modulation of both E complex and other splicing factor (e.g., U2) binding kinetics may be important to either enhance or suppress redefinition events and permit proper 5′ SS and BS selection.

## U1 binding is tuned at both weak and strong splice sites

While it has often been noted that alternative splicing correlates with low complementarity between the 5′ SS and the U1 snRNA, it is less clear what factors lead to discrimination between splice sites of equal strength (*Roca et al., 2013*). This discrimination could arise from *trans*-acting factors tuning U1 binding at particular locations to either stabilize or destabilize snRNP association. Our data indicate that yeast ECPs function in this capacity to facilitate U1 recruitment to pre-mRNAs containing weak splice sites (*Figure 1D*), modulate U1 interactions occurring at consensus splice sites (*Figure 2*), and destabilize U1 binding if excessive complementarity is present (*Figures 1*, *2* and *5*). Tuning of U1 interactions is thus a general feature of both weak and strong splice sites, and the lifetime of a U1/pre-mRNA complex may not be apparent from base pairing potential alone. In our experiments, we focused on substrates containing only a single 5′ SS; however, it is possible that tuning may also change how two or more potential 5′ SS compete with one another within a single transcript.

Extensive analysis has revealed a rich network of genetic and physical interactions between the CBC, BBP/Mud2, and both the proteins and snRNA of U1 (*Chang et al., 2012*; *Qiu et al., 2012*; *Schwer and Shuman, 2014*; *Agarwal et al., 2016*; *Schwer and Shuman, 2015*; *Schwer et al., 2013*; *Zhang and Rosbash, 1999*). While the contribution RNA conformation plays in promoting U1 release is not yet known, the single molecule and genetic data suggest U1 tuning is occurring through physical connections between components of the E complex. The measured E complex lifetimes (*Figure 3E*) allow us to define that any such physical interactions resulting in a cross-intron bridging complex (*Abovich and Rosbash, 1997*) between U1 at the 5′ SS and BBP/Mud2 at the BS are transient and likely to dissociate with a rate constant of ~0.4 min$^{-1}$ or greater.

Tuning U1 interactions with cognate RNAs may be critical for processes other than spliceosome assembly including those that are splicing-independent such as telescripting (*Berg et al., 2012*). While stable association of U1 promotes assembly, failure to release U1 during spliceosome activation inhibits splicing since the U6 snRNA must also pair to the 5′ SS (*Chiou et al., 2013*; *Staley and Guthrie, 1999*). A two-step model for U1 binding (*Figure 6A*) permits a reversible change in affinity that could be used to decrease U1 stability during activation. The more weakly-bound U1 complex may then facilitate U1/U6 exchange. Kinetic data obtained with the L13F and D36A mutants of Yhc1 support this idea (*Figure 4*). These mutations bypass (L13F) (*Chen et al., 2001*) or reinforce (D36A) (*Schwer and Shuman, 2014*) the need for the Prp28 ATPase to promote activation, and U1's

containing these mutations would be predicted to associate with the 5′ SS less (L13F) or more (D36A) strongly than WT. Indeed, we found this prediction to be true but the effects were modulated by ECPs. Other components of the spliceosome known to interact with U1 [(e.g., Prp8; (*Li et al., 2013*)] may function similarly to tune U1 binding during later stages of splicing.

## Tuning changes how transcripts compete for U1

Genetic experiments have elucidated functional redundancy between the CBC and BBP/Mud2 with respect to their ability to complement mutations in U1 (*Qiu et al., 2012*; *Chang et al., 2012*; *Schwer and Shuman, 2014*; *Agarwal et al., 2016*; *Schwer and Shuman, 2015*; *Schwer et al., 2013*). We were also able to observe some functional redundancy for these factors at the single molecule level, as both ECPs could extend U1 lifetimes (*Figures 1* and *2*). Whether or not both ECPs accomplish this by a similar mechanism is not yet clear. However, the BS exerts an additional and unexpected effect on U1 by increasing the relative abundance ($A_2$) of long-lived complexes. We rationalize this by BBP/Mud2 bound at the BS impacting the transition between initial and stable U1 binding in the two-step model (*Figure 6A*). This model could also explain differences in genetic lethality observed when snRNA mutations are combined with Mud2Δ or Cbp20 mutations, with the former more often being lethal than the latter (*Schwer et al., 2013*). While BBP/Mud2 and CBC each help to anchor U1 to transcripts, their molecular influences on U1 are distinct.

One consequence of this observation is that transcripts destined for splicing (i.e., those containing introns) are more competitive for U1 compared to those which only contain 5′ SS sequences or only bound by ECPs (*Figure 6B*). We propose that CBC initially facilitates sequence-independent U1 recruitment to transcripts, which biases U1 occupancy towards Pol II transcription products that may contain introns. Since the 5′ end of the snRNA is not required for transient, ECP-dependent interactions (*Figure 1—figure supplement 3*), initial recruitment may not involve pairing between the snRNP and transcript. U1 could then subsequently engage with a cognate 5′ SS sequence, at which point it may or may not switch into a more stably bound conformation. If the transcript also contains a downstream BS bound by BBP/Mud2, this promotes recruitment as well as the switch to stable binding. The features of this model are consistent with in vivo data obtained in both yeast and humans which have detected splicing-independent recruitment of U1 and the influences of the CBC and BS on U1 occupancy (*Görnemann et al., 2005*; *Lacadie and Rosbash, 2005*; *Patel et al., 2007*; *Spiluttini et al., 2010*). Notably, the two-step binding mechanism (*Figure 6A*), in which BBP/Mud2 influences U1's transition between initial and stable complex formation, results in a coupling of 5′ SS and BS recognition. Stable U1 binding can be associated with multiple landmarks found on the transcript including the cap, 5′ SS, and BS—all indicators of an intron-containing pre-mRNA.

## Conclusion

Rapid identification of specific nucleic acid sequences is a challenge faced by many cellular RNPs including the ribosome, the RNA-induced silencing complex (RISC), and bacterial CRISPR machineries in addition to spliceosome snRNPs. While mechanisms of ribosome scanning remain to be elucidated (*Archer et al., 2016*; *Hinnebusch, 2014*), recent structural and single-molecule data indicate that both Cas9 and Argonaute share common strategies for finding particular sequences of DNA or RNA (*Chandradoss et al., 2015*; *Salomon et al., 2015*; *Sternberg et al., 2014*). These RNP machineries utilize 'seed' or PAM sequences to facilitate rapid scanning of nucleic acids in a step distinct from that of formation of extended base pairing interactions. These two-step mechanisms enable efficient target identification and likely also limit the lifetimes of non-productive interactions. Our data support a similar two-step mechanism for U1 snRNP recruitment to 5′ SS in which short-lived interactions give way to longer-lived complexes. Whether or not U1 also contains a 'seed' region that nucleates binding is not yet clear; however, the U1-C/Yhc1 protein is positioned to potentially participate in such a process. This could explain the origin of our observed kinetics as well as provide U1 an efficient means of searching for potential 5′ SS.

## Materials and methods

### Pre-mRNA preparation

Capped [$^{32}$P]-labeled RP51A pre-mRNAs for in vitro splicing and trace [$^{32}$P]-labeled RP51A substrates for single molecule assays were made by in vitro transcription with T7 RNA polymerase in the presence of [α-$^{32}$P] UTP and G(5')ppp(5')G RNA cap analog (NEB). Trace [$^{32}$P]-labeled RP51A was fluorescently labeled by splinted ligation with a biotinylated 2'-O-methyl oligonucleotide derivatized with a single Alexa Fluor 488 (Alexa488, ThermoFisher Scientific) or Cy5 (GE Life Sciences) fluorophore as previously described.

### Preparation of yeast strains

Yeast strains containing fast SNAP (SNAP$_f$) (*Sun et al., 2011*) tags on the U1 snRNP proteins Prp40 and Snp1 or BBP (*Supplementary file 4*) were prepared by homologous recombination as previously described (*Hoskins et al., 2016*). The SNAP$_f$-tagged Yhc1 deletion strain and Yhc1 plasmids (*Supplementary file 4* and *6*) were kind gifts of Dr. Magda Konarska and Dr. Beate Schwer, respectively. The triple-labeled strain containing DHFR tags on Prp40 and Snp1 and a SNAP$_f$ tag on BBP was prepared from strain yAAH3 (*Hoskins et al., 2011*) and by SNAP$_f$ tagging the Msl5 (BBP) gene by homologous recombination as previously described. Alleles of Yhc1 were exchanged by plasmid shuffling and selection for loss of the WT allele on dropout plates containing 1 mg/mL 5-FOA.

### Preparation and labeling of yeast whole-cell splicing extracts

Yeast whole cell extract (WCE) was prepared as previously described (*Crawford et al., 2007*). SNAP$_f$-tagged proteins were labeled by incubation of the lysate for 30 min at room temperature with the fluorophore (e.g., benzylguanine-Dy549/SNAP-Surface 549, New England Biolabs) before gel filtration. A fluorophore concentration of 1.1 μM was used to label SNAP$_f$ tags.

### Visualization of SNAP$_f$-tagged proteins in SDS-PAGE gels

SNAP$_f$-tagged proteins derivatized with fluorophores were visualized by denaturing polyacrylamide gel electrophoresis (SDS-PAGE) followed by imaging fluorescence on a LAS 4000 or Typhoon fluorescence imager (GE Life Sciences).

### In vitro splicing assays

Splicing assays were carried out as previously described using 40% WCE and 0.2–0.5 nM substrate (*Crawford et al., 2007*). Splicing assays with WCE used in single molecule experiments also contained the oxygen scavengers and triplet quenchers described below. [$^{32}$P]-labeled RNAs were visualized by denaturing PAGE followed by phosphorimaging. Data were analyzed using ImageQuant software (GE Lifesciences).

### Commitment complex assays

Commitment complex assays were performed as previously described with the following modifications (*Seraphin and Rosbash, 1989*). 50 μL standard splicing reactions containing 40% WCE were depleted of ATP by the addition of 2 mM glucose and incubating at 25°C for 20 min. [$^{32}$P]-labeled RP51A was prepared from a template truncated at the DdeI site within the *RP51A* gene. The RNA (0.5 nM) was then added to the reaction and incubated for 20 min at 16°C. 10 μL of the reaction was added to 10 μL of ice cold buffer R [2 mM MgOAc, 50 mM HEPES pH 7.5, 1 mg/mL tRNA, 50-fold molar excess of cold RP51A pre-mRNA; (*Price et al., 2014*)] and incubated for 10 min on ice. 5 μL of loading dye (2.5x TBE, 50% v/v glycerol, 0.3% w/v bromophenol blue, and 0.3% w/v xylene cyanol dyes) was added to the solution before it was loaded on a 0.5x TBE, 3% 60:1 acrylamide, 0.5% w/v agarose, 2% v/v glycerol gel. The 26 cm gel was run in 0.5x TBE at 12V for 24 hr at 4°C. The gel was then dried before phosphorimaging.

### U1 ablation

Oligonucleotide directed RNase H digestion of the 5' end of the U1 snRNA was carried out by addition of 6.5 μg of a DNA oligo complimentary to the 5' end of the U1 snRNA (5'-CTTAAGGTAAG-TAT-3') and 0.048 U/μL RNase H (Invitrogen) to a 100 μL splicing reaction and incubating at 30°C for

30 min (*Du and Rosbash, 2001*). Part of the reaction was quenched by adding 20 µL of the reaction solution to 180 µL of splicing dilution buffer (0.1 M Tris pH 7.5, 1% w/v SDS, 0.3 M NaOAc, 150 mM NaCl, 1 mM EDTA) and placed on ice. The remaining reaction solution was immediately used for single molecule assays after the addition of triplet state quenchers and oxygen scavengers.

## Primer extension

Primers complementary to nucleotides 114–135 (5'-GACCAAGGAGGTTGCATCAATG-3') of the U1 snRNA and nucleotides 100–121 of the (5'-GCCAAAAAATGTGTATTGTAA-3') of the U2 snRNA were end labeled using [γ-$^{32}$P] ATP (PerkinElmer). Total yeast RNA was isolated by phenol extraction and ethanol precipitation from the quenched U1 ablation reaction and resuspended in 10 µL of water. The extracted RNA (1 µL) was added to 1.5 µL of oligo mix (0.4 pmol/µL [γ-$^{32}$P] U1 oligo, 0.4 pmol/µL [γ-$^{32}$P] U2 oligo, 1.25M KCl, and 50 mM Tris pH 8.0) and placed on ice for 3 min. Primers were annealed by heating the mixture to 90°C and then cooling on ice for 3 min each. The solution was then heated to 45°C for 5 min before addition of 6.5 µL of RT mix [150 mM Tris pH 8.0, 50 mM MgCl2, 50 mM DTT, 2.5 mM dNTPs, 0.3 µL SuperScript III Reverse Transcriptase (Invitrogen)]. Primer extension products were analyzed by 6% denaturing polyacrylamide gel electrophoresis. The gel was then dried before phosphorimaging.

## Single-molecule experiments

Cleaning of slides and coverslips and assembly and passivation of flow cell chambers were carried out as previously described (*Crawford et al., 2007*). Two-color CoSMoS assays were carried out in splicing buffer with the addition of PCD/PCA as an oxygen scavenging system and trolox as previously described. Three-color CoSMoS experiments additionally included cyclooctatetraene, propyl gallate, and 4-nitrobenzylacohol as triplet quenchers as previously described (*Hoskins et al., 2011*). ATP was depleted from WCE by addition of 2 mM glucose prior to the experiment. Identical U1 and BBP binding kinetics were observed in the absence and presence of additional exogenous, purified hexokinase [added to final concentration of 5 U/mL, Roche 11426362001 (1500 U/mL, ~450 U/mg protein); *Supplementary file 2* and *3*]. Therefore, only glucose addition and the endogenous hexokinase activity present in the extract was used to deplete ATP.

A custom-built, objective-based micromirror TIRFM (*Larson et al., 2014*) was used to image the interactions between fluorescently labeled, surface-tethered RNAs and fluorescently labeled U1 or BBP molecules. Two-color and three-color CoSMoS experiments were carried out as previously described with the following modifications (*Hoskins et al., 2011*). Laser powers were set between 200 and 500 µW for both the 532 and 633 nM lasers in two- and three-color experiments and 1.5 mW for the 488 nm laser in three-color experiments. U1 snRNPs and BBP were imaged with a 1 s exposure at 5 s intervals between each frame. Surface immobilized RNAs were imaged with a 1 s exposure every 60 frames, except in three-color experiments during which RNAs were imaged only at the beginning and end of the experiment. Photobleaching analysis was carried out as described by varying the laser power in each experiment and measuring U1 and BBP dwell times (*Hoskins et al., 2016*). Under these imaging conditions, the contribution of photobleaching to U1 and BBP the fitted parameters describing the distribution of dwell times was minimal.

Drift correction was achieved by tracking changes in position of the surface tethered RP51A throughout the durations of the experiment and was seldom more than one pixel in any direction during the duration of each experiment. Auto focusing was carried out using a 785 nm laser immediately before imaging of the immobilized RNAs in all experiments. Mapping between the <635 nm and >635 fluorescence fields of view (FOV) was achieved using a reference data set generated using fluorescent beads which emit light at multiple wavelengths.

## Data analysis

Data analyses were done as previously described (*Hoskins et al., 2011*; *Shcherbakova et al., 2013*). In brief, fluorescence signal from surface tethered pre-mRNAs were used to select areas of interest (AOIs). AOIs were mapped from the >635 nm FOV to the <636 nm FOV and pixel intensity was integrated for each AOI using custom MATLAB software. Peaks in the fluorescence intensity from the <635 nm FOV were manually inspected to confirm the presence of a colocalized spot in the AOI.

The distributions of observed dwell times for each subcomplex were displayed by constructing probability density plots in which the dwell times were binned and each bin divided by the product of the bin width and total number of events. Error bars for each bin were calculated as the error of a binomial distribution as previously described (*Hoskins et al., 2011*). Distributions described by one or two exponential terms were fit by maximum likelihood methods to exponential probability density functions as previously described using *Equations 1 or 2*, respectively (*Hoskins et al., 2011*). In all equations, $t_m$ represents the time between consecutive frames; $t_{max}$ represents the duration of the experiment (30 min); $A_1$ and $A_2$ the fitted amplitudes; and $\tau_1$ and $\tau_2$ represent the fitted parameters. Errors in the fit parameter were determined by bootstrapping 1000 random samples of the data and determining the standard deviation of the resultant values.

$$\left[\left(A_1 \cdot \left(e^{-\frac{t_m}{\tau_1}} - e^{-\frac{t_{max}}{\tau_1}}\right)\right)\right]^{-1} \cdot \left[\frac{A_1}{\tau_1} e^{\frac{-t}{\tau_1}}\right] \tag{1}$$

$$\left[\left(A_1 \cdot \left(e^{-\frac{t_m}{\tau_1}} - e^{-\frac{t_{max}}{\tau_1}}\right)\right) + \left((1-A_1) \cdot \left(e^{-\frac{t_m}{\tau_1}} - e^{-\frac{t_{max}}{\tau_1}}\right)\right)\right]^{-1} \left[\frac{A_1}{\tau_1} e^{\frac{-t}{\tau_1}} + \frac{1-A_2}{\tau_2} e^{\frac{-t}{\tau_2}}\right] \tag{2}$$

## Acknowledgements

We thank Magda Konarska and Beate Schwer for strains and plasmids. This work was supported by the National Institutes of Health [R00 GM086471, R01 GM112735], a Shaw Scientist Award, a Beckman Young Investigator Award, startup funding from the University of Wisconsin-Madison, Wisconsin Alumni Research Foundation (WARF), and the Department of Biochemistry

## Additional information

### Funding

| Funder | Grant reference number | Author |
|---|---|---|
| National Institute of General Medical Sciences | R01 GM112735 | Joshua Donald Larson<br>Aaron A Hoskins |
| Greater Milwaukee Foundation | | Joshua Donald Larson<br>Aaron A Hoskins |
| Arnold and Mabel Beckman Foundation | | Joshua Donald Larson<br>Aaron A Hoskins |
| Wisconsin Alumni Research Foundation | | Joshua Donald Larson<br>Aaron A Hoskins |
| National Institutes of Health | R00 GM086471 | Joshua Donald Larson<br>Aaron A Hoskins |

The funders had no role in study design, data collection and interpretation, or the decision to submit the work for publication.

### Author contributions

Joshua Donald Larson, Conceptualization, Data curation, Formal analysis, Writing—original draft, Writing—review and editing; Aaron A Hoskins, Conceptualization, Supervision, Funding acquisition, Validation, Investigation, Methodology, Writing—original draft, Project administration, Writing—review and editing

### Author ORCIDs

Aaron A Hoskins http://orcid.org/0000-0002-9777-519X

### Decision letter and Author response

Decision letter https://doi.org/10.7554/eLife.27592.033
Author response https://doi.org/10.7554/eLife.27592.034

# Additional files

## Supplementary files

• Supplementary file 1. Sequences of RNAs used in these experiments.
DOI: https://doi.org/10.7554/eLife.27592.026

• Supplementary file 2. Fit Parameters for the Distribution of U1 Lifetimes Under Various Conditions
DOI: https://doi.org/10.7554/eLife.27592.027

• Supplementary file 3. Fit Parameters for the Distribution of BBP Lifetimes on RNAs Containing or Lacking the BS
DOI: https://doi.org/10.7554/eLife.27592.028

• Supplementary file 4. Yeast strains used in these experiments.
DOI: https://doi.org/10.7554/eLife.27592.029

• Supplementary file 5. Fit Parameters for the Distribution of Lifetimes of E Complexes Containing Colocalized U1 and BBP
DOI: https://doi.org/10.7554/eLife.27592.030

• Supplementary file 6. Plasmids for Yeast Protein Expression and Encoding DNA Templates for RNA Substrates
DOI: https://doi.org/10.7554/eLife.27592.031

• Transparent reporting form
DOI: https://doi.org/10.7554/eLife.27592.032

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
