## [Decision Letter]

Thank you for submitting your article "Dynamics and Consequences of Spliceosome E Complex Formation" for consideration by *eLife*. Your article has been favorably evaluated by Michael Marletta (Senior Editor) and three reviewers, one of whom, Jonathan P Staley (Reviewer #1), is a member of our Board of Reviewing Editors. Your article has been reviewed by 3 peer reviewers, and the evaluation has been overseen by a Reviewing Editor and Michael Marletta as the Senior Editor.

The reviewers have discussed the reviews with one another and the Reviewing Editor has drafted this decision to help you prepare a revised submission.

Summary:

This manuscript describes a high resolution dissection of the binding of the U1 snRNP to a 5' splice site and the impact of pre-mRNA features and splicing co-factors on this binding. To achieve this goal, the authors leverage the illuminating single-molecule approach, termed CoSMOS, in the context of yeast extract. A major conclusion is that U1 binds in two distinct states, one short-lived and one-long lived, which the authors interpret to indicate a pathway of binding in which a less stable, initial state transitions to a more stable state, a pathway that parallels two-step pathways for binding of Cas9 and Ago RNPs to nucleic acid. Further, the cap, presumably through the cab binding complex (CBC), and the branch site, through the branch site binding protein (BBP), both help recruit U1 snRNP to pre-mRNA, independent of a 5' splice site, and both can increase the stability of the more stable state; interestingly, however, the branch site but not the cap can increase the probability that the stable state forms, suggesting that the branch site promotes the transition to the stable state, in the authors' model. Interestingly, a mutation in U1-C impacts the lifetime of the less stable state, implicating a role for this protein at this presumably earlier, transient stage. Additionally, simultaneous dynamic analysis of U1 and BBP binding to pre-mRNA reveals that the 5' splice site and branch site can be redefined after initial complex formation on pre-mRNA, indicating that this "commitment complex", as originally described, is not committed. This manuscript will be of interest to a broad audience, including the single molecule, nucleic acid binding, and splicing communities. Technically, there is little to criticize in this study, in which, beyond the two-state model, a number of quantitative observations come together to change, clarify, and deepen the way we view 5' splice site recognition by the U1 snRNP. Still, the reviewers have several important questions that need to be answered before publication can be considered further.

Essential revisions:

1) The model of weak U1 binding to 5'SS followed by strong binding is not unambiguously supported by the data. The weak and strong binding events observed in the experiments described here could arise from completely independent binding events, instead of sequential ones. Fortunately, the proposed model makes testable predictions, which the authors need to test. For example, the model predicts that varying the concentration of the U1 snRNP would impact the amplitude to τ_1_ but not the amplitude of τ_2_. Of course, these experiments are performed in extract, but testing the impact of increasing U1 snRNP would be quite informative as to the relevance of the model.

2) Importantly, the functional relevance of the different states is unclear. For example, in Figure 1, it is shown that the probability of "stable binding" (indicated by A_2_) is at background levels for RNA 2, yet RNA 2 is apparently spliced as well as RNA 3 (Figure 1—figure – supplement 2) – at least by eye, as no quantitation is given. Perhaps the splicing of RNA 2 is less efficient or kinetically slower than RNA 3. This could be tested by quantifying and/or performing a time course of splicing, which should be done. If the splicing of RNA 2 and 3 really are indistinguishable, then more consideration needs to be given as to whether the two-state model is functionally impactful and whether the measured parameters matter for the ultimate activity of the spliceosome. For example, would the parameters matter in the context of competing 5' splice sites?

3) Different conclusions seem to result from either mutation of the branch site or observing BBP directly. Specifically, the data in Figure 2 imply that either the CBC or BBP are sufficient to stabilize τ_2_ (i.e., the loss of the branch site is insufficient to alter τ_2_). However, in Figure 3, BBP, presumably in the presence of CBC, correlates with longer U1 dwell times. How can these views be reconciled? Does this imply that BBP impacts U1 stability at a 5' splice site independent of the branch site? Indeed, Gornemann et al. (2005) have shown that BBP is recruited to a transcript by ChIP at the same time as the U1 snRNP after the 5' splice site has been transcribed but before the branch site has been transcribed. A simple experiment to resolve this apparent paradox would be to observe E complex formation, via U1 and BBP, on a substrate that lacks a branch site. Minimally, this discrepancy needs to be acknowledged.

4) How can the authors be certain that residual ATP is not sufficient to explain some of the dynamics? It seems critical to test this presumption by incubation of the extract with glucose/hexokinase.

---

## [Author Response]

Essential revisions:1) The model of weak U1 binding to 5'SS followed by strong binding is not unambiguously supported by the data. The weak and strong binding events observed in the experiments described here could arise from completely independent binding events, instead of sequential ones. Fortunately, the proposed model makes testable predictions, which the authors need to test. For example, the model predicts that varying the concentration of the U1 snRNP would impact the amplitude to τ_1_ but not the amplitude of τ_2_. Of course, these experiments are performed in extract, but testing the impact of increasing U1 snRNP would be quite informative as to the relevance of the model.

We believe that our data support two-step binding by U1 at the 5’SS. We have edited this section of the manuscript (subsection “Properties of the U1/5' SS Interaction”) to clarify our evidence for this mechanism in several important ways. First, we have included a new supplemental figure (Figure 6—figure supplement 1) which describes the most likely alternate mechanism (two distinct and independent binding pathways) that could account for the data. This allows us to frame our discussion of two-step binding more easily in ways that either support two-step binding or do not support independent binding events. Second, we explain in further detail the consequences of observing coupled changes in lifetimes and amplitudes (for example, increases in τ_2_ accompanied by increases in A_2_). These observations are likely incompatible with independent binding since it is less clear how changing the stability of a particular complex would alter the abundance of another unless they are kinetically coupled.

We agree with the reviewer’s comment that our model makes testable predictions. However, obtaining additional kinetic parameters (such as on-rates) for U1 in cell extract is very difficult since it is not apparent how the concentration of endogenous U1 can be easily changed while keeping the overall amount of cell extract constant. Such experiments, if possible, are beyond the scope of the work presented here. While it is difficult to unambiguously prove a mechanism for any biomolecule, we believe our data support two-step binding and are less compatible with the most likely alternative.

2) Importantly, the functional relevance of the different states is unclear. For example, in Figure 1, it is shown that the probability of "stable binding" (indicated by A_2_) is at background levels for RNA 2, yet RNA 2 is apparently spliced as well as RNA 3 (Figure 1—figure supplement 2) – at least by eye, as no quantitation is given. Perhaps the splicing of RNA 2 is less efficient or kinetically slower than RNA 3. This could be tested by quantifying and/or performing a time course of splicing, which should be done. If the splicing of RNA 2 and 3 really are indistinguishable, then more consideration needs to be given as to whether the two-state model is functionally impactful and whether the measured parameters matter for the ultimate activity of the spliceosome. For example, would the parameters matter in the context of competing 5' splice sites?

The reviewer raises several interesting points. First, we have quantified the splicing of RNA 2 and RNA 3 in Figure 1—figure supplement 2, as the reviewer suggested, and have included the data in that figure legend. The 5’SS cleavage (1^st^ step) efficiency for RNA 2 (which contains the weak 5’ SS) is only 8% vs. 25% for RNA 3 (which contains the consensus, RP51A 5’ SS). The exon ligation (2^nd^ step) efficiencies are 4 and 16% for RNA 2 and RNA 3, respectively. This is consistent with weaker 5’SS being spliced less efficiently in vitro and as well as with previous analysis of 5’SS mutants using a variety of assays (for example, Pikielny and Rosbash, Cell, 1985 and Seraphin and Rosbash, EMBO J., 1991). Thus, RNA 2 both binds U1 more weakly and splices less efficiently.

In terms of ultimate activity of the spliceosome, we have included new text discussing the functional relevance of U1 interactions (subsection “The Spliceosome E Complex is Dynamic”). Since the 5’ SS is recognized sequentially by multiple spliceosome components, our parameters may matter for splicing more so when U1 binding is limiting and less so under other conditions (for example, when U2/BS pairing is limiting). Finally, we previously included text describing how U1C mutants that alter dependence of the spliceosome on Prp28 activity also change our observed binding events in predicted ways (excepting the influence of ECPs), suggesting that these different states are functionally relevant for Prp28-dependent steps that occur after initial U1 binding (subsection “U1 Binding is Tuned at Both Weak and Strong Splice Sites”).

The reviewer also asks how would our parameters matter in the context of competing 5’ splice sites. This is an interesting idea, and we have included a statement in which we speculate on the significance of our results with respect to 5’SS competition (subsection “U1 Binding is Tuned at Both Weak and Strong Splice Sites”, last paragraph).

3) Different conclusions seem to result from either mutation of the branch site or observing BBP directly. Specifically, the data in Figure 2 imply that either the CBC or BBP are sufficient to stabilize τ_2_ (i.e., the loss of the branch site is insufficient to alter τ_2_). However, in Figure 3, BBP, presumably in the presence of CBC, correlates with longer U1 dwell times. How can these views be reconciled? Does this imply that BBP impacts U1 stability at a 5' splice site independent of the branch site? Indeed, Gornemann et al. (2005) have shown that BBP is recruited to a transcript by ChIP at the same time as the U1 snRNP after the 5' splice site has been transcribed but before the branch site has been transcribed. A simple experiment to resolve this apparent paradox would be to observe E complex formation, via U1 and BBP, on a substrate that lacks a branch site. Minimally, this discrepancy needs to be acknowledged.

The reviewer is comparing kinetic parameters obtained from maximum likelihood fitting of the dwell times (Figure 2) with an average of the observed dwell times (Figure 3). For multi-exponential kinetics, these two types of analysis cannot be directly compared, and we apologize for the confusion. We wish to retain the analysis shown in Figure 3 since it presents directly that co-binding of U1 and BBP results in longer U1 lifetimes.

To clarify this point, we have expanded the discussion of this data and have included comparisons with average dwell times derived from data shown in Figure 2 (subsection “BBP Associates with the Longest-Lived U1/Pre-mRNA Complexes”, last paragraph). These data confirm that the CBC alone cannot account for the extension of U1 lifetimes observed during colocalization with BBP and that the CBC was likely influencing U1 lifetimes when BBP was not bound.

The reviewer also wonders about BBP binding to a transcript in the presence of a 5’ SS but lacking a BS. We had previously included such an experiment (Figure 2—figure supplement 1, RNA 7) and failed to observe any BBP binding in the absence of the BS and pseudo-BS, suggesting U1 alone cannot recruit BBP. This is in contrast to the model proposed by Gornemann et al. It is possible that our data and those of Gornemann et al. can be reconciled by transient binding of BBP to pseudo-BS found within the DBP2 transcript analyzed by Gornemann et al. DBP2 contains a long intron with possible pseudo-BS sequences (for example, CCCUAAC in DBP2 located near the 5’SS). Alternatively, cross-linking experiments are irreversible and may capture very short-lived or rare interactions that we are unable to observe. We have included a reference and discussion of the work of Gornemann et al. in the third paragraph of the subsection “The RNA Cap and BS Facilitate U1 Recruitment in the Absence of a Strong 5ʹ SS”.

In sum, we do not believe there is an apparent discrepancy in these data and that the additional analysis and textual modifications clarify the points addressed above.

4) How can the authors be certain that residual ATP is not sufficient to explain some of the dynamics? It seems critical to test this presumption by incubation of the extract with glucose/hexokinase.

All of our experiments were carried out in the absence of ATP by simultaneously not adding ATP to the extract and using glucose to deplete endogenous ATP via the endogenous yeast hexokinase. We have tested whether or not the endogenous hexokinase was sufficient by carrying out new experiments suggested by the reviewer in which U1 and BBP lifetimes were measured after incubation of the extract with both glucose and additional, purified yeast hexokinase. Under these new conditions, U1 and BBP dynamics and lifetimes were indistinguishable from those measured using glucose-only depletion (Supplementary file 2 and Supplementary file 3, subsection “Single-Molecule Experiments”, first paragraph). Thus, we feel that residual ATP is unlikely to explain the observed dynamics.